# Prognostic Factors for Overall Survival in Patients with HCV-Related HCC Undergoing Molecular Targeted Therapies: Beyond a Sustained Virological Response

**DOI:** 10.3390/cancers14194850

**Published:** 2022-10-04

**Authors:** Yasunori Minami, Tomoko Aoki, Hirokazu Chishina, Masahiro Takita, Satoru Hagiwara, Hiroshi Ida, Kazuomi Ueshima, Naoshi Nishida, Masatoshi Kudo

**Affiliations:** Department of Gastroenterology and Hepatology, Faculty of Medicine, Kindai University, 377-2 Ohno-Higashi Osaka-Sayama, Osaka 589-8511, Japan

**Keywords:** ALBI grade, BCLC stage, FIB-4 index, hepatocellular carcinoma, molecular targeted therapy, prognostic factor, overall survival, sustained virological response

## Abstract

**Simple Summary:**

The eradication of the hepatitis C virus (HCV) has had a significant impact on the management of patients with HCV-related hepatocellular carcinoma (HCC). However, its eradication has not completely resolved survival issues in patients with HCV-related HCC. Therefore, the present study investigated prognostic factors for survival in patients with HCV-related HCC undergoing molecular targeted therapies. In total, 359 HCC patients treated with first-line chemotherapy were enrolled. The median follow-up duration was 16.0 months (range, 1.0–115.7) and the median duration of first-line systemic therapy was 3.7 months (range, 0.7–86.9). The achievement of a sustained virological response (SVR) (*p* < 0.001), albumin-bilirubin (ALBI) grade II/III (*p* < 0.001), Barcelona Clinic Liver Cancer (BCLC) stage C (*p*  =  0.005), extrahepatic spread (*p* < 0.001), baseline AFP level ≥ 90 (*p* = 0.038), baseline DCP level ≥ 500 (*p* < 0.001), and a fibrosis-4 (FIB-4) index ≥ 4 (*p*  =  0.003) were identified as independent prognostic factors for overall survival.

**Abstract:**

Background: The treatment of the hepatitis C virus (HCV) has reduced the risk of hepatocellular carcinoma (HCC)-related mortality. Many patients with advanced HCC have achieved longer survival through systemic chemotherapy. However, survivors of HCC may develop liver cancer during and after treatment. Therefore, the present study investigated prognostic factors for survival in patients with HCV-related HCC in the new era of molecular targeted therapy. Methods: A total of 359 patients with HCV-related HCC treated with first-line chemotherapy were reviewed. A Cox proportional hazards model and Kaplan–Meier curve were used to identify prognostic factors associated with survival outcomes. Results: The median follow-up duration was 16.0 months (range, 1.0–115.7) and the median duration of first-line systemic therapy was 3.73 months (range, 0.7–86.9). The achievement of a sustained virological response (SVR) (*p*  <  0.001), albumin–bilirubin (ALBI) grade II/III (*p*  <  0.001), Barcelona Clinic Liver Cancer (BCLC) stage C (*p*  =  0.005), extrahepatic spread (*p* < 0.001), baseline AFP (alpha-fetoprotein) level ≥ 90 (*p* = 0.038), baseline DCP (des-γ-carboxy prothrombin) level ≥ 500 (*p* < 0.001), and a fibrosis-4 (FIB-4) index ≥ 4 (*p*  =  0.003) were identified as prognostic factors for overall survival. Conclusions: The achievement of SVR was most strongly associated with overall survival. Other factors, such as the BCLC stage, extrahepatic spread, baseline tumor marker (AFP/DCP) levels, ALBI grade, and FIB-4 index need to be considered in the management of patients with HCV-related HCC.

## 1. Introduction

Hepatocellular carcinoma (HCC) accounts for 90% of primary liver cancers, which is the sixth most common cancer worldwide and the third major cause of cancer-related death [1]. The estimated number of new cases of liver cancer and deaths in 2020 was approximately 906,000 and 830,000, respectively [1], and more than one million patients will suffer from liver cancer per year by 2025 [2]. The hepatitis C virus (HCV) is the major causative agent of HCC, mainly through indirect pathways: chronic inflammation, cell deaths, and proliferation. Approximately 3% of the world population is infected with HCV, and the severe consequences of viral infection makes HCV one of the most pressing emergencies worldwide [3].

Advances have been achieved in the management of HCC among patients with chronic liver disease; however, clinical outcomes depend on both the cancer stage and the degree of liver impairment. With the development and introduction of early HCC detection techniques, peri-operative management, and technical procedures, locoregional therapies, including surgical resection, thermal ablation, and transcatheter arterial chemoembolization have increased survival [4,5]. Systemic chemotherapy drugs, such as multi-molecular targeting agents, monoclonal antibodies, and immune-checkpoint inhibitors (ICIs) have also been recently developed. Approximately 50–60% of patients achieved longer survival through systemic therapies, particularly in the advanced stages of HCC [3,4,5,6,7,8,9]. Cirrhosis represents the final stage for a wide variety of chronic liver diseases and may cause complications, including esophageal varices and ascites. Although acute variceal bleeding is a fatal emergency in cirrhotic patients, variceal bleeding has been markedly reduced via prophylactic therapy to prevent future bleeds [10]. The treatment of ascites has been shown to improve the quality of life of patients and decrease the risk of developing spontaneous bacterial peritonitis [11]. Most importantly, the efficacy of HCV treatments has markedly improved over the past decade, even though HCV infection is a major cause of HCC worldwide. The initial treatment, consisting of interferon, has been replaced by direct-acting antivirals, which have achieved sustained virological response (SVR) rates higher than 95% regardless of the treatment regimen, presence of cirrhosis, or HCV subtype [12]. Viral eradication is essential to preventing disease progression and reduces hepatitis C-related mortality and morbidity [13]. Previous studies suggested that the elimination of HCV has had a significant impact on survival in HCC patients [14,15,16]. However, not all survival issues have been resolved, and therefore we need to focus on other prognostic factors. Furthermore, targeted therapy has been heralded by many as the holy grail of treatment for advanced HCC. With the development of systemic chemotherapy, the survival time of patients with HCC has increased, and a new era of treatment for advanced HCC has arrived, with exciting data presented for molecular targeted therapies. However, cancer survivors who received systemic chemotherapy may face unique challenges during and after treatment. The problem that HCC survivors face in managing care are common due to the complex association between HCC and chronic liver disease. Therefore, the present study investigated the prognostic factors for survival in patients with HCV-related HCC in the new era of molecular targeted therapy.

## 2. Materials and Methods

### 2.1. Patients

This was a single-institute, retrospective analysis of HCC patients treated with first-line molecular targeted therapies between May 2009 and March 2021. Consecutive patients were treated with sorafenib as first-line systemic therapy for unresectable HCC from May 2009, lenvatinib from August 2019, and the combination of atezolizumab and bevacizumab (ATZ/BV) from October 2020 onwards. HCC was diagnosed based on histological findings or radiological modalities according to American Association for the Study of Liver Disease clinical guidelines after the detection of arterial hyperenhancement and washout on delayed-phase imaging [17,18]. Inclusion criteria were as follows: an age of 20–95 years, an Eastern Cooperative Oncology Group performance status 0–1, the presence of intrahepatic tumors that influence the prognosis of patients regardless of extrahepatic metastasis, adequate organ function, and a Child–Pugh score 5–7. Exclusion criteria were hepatitis B virus infection, a negative test result for hepatitis C, prior systemic anticancer therapy and adjuvant post operative chemotherapy, BCLC stage D, pleural effusion or ascites refractory to treatment, hepatic encephalopathy, severe and active concomitant malignancy, and an unsatisfactory general condition.

The Institutional Review Board of Kindai University Hospital (IRB No. 27-136) reviewed and approved the present study. Since this was a retrospective analysis, the need for informed consent was waived.

### 2.2. Treatment Regimens

Oral treatment with sorafenib (Nexavar, Bayer HealthCare/Onyx Pharmaceuticals) was initiated at a dose of 800 mg (400 mg twice daily). Patients were orally administered lenvatinib (Lenvima, Eisai Co., Ltd., Tokyo, Japan) at the following doses: 8 mg/day for patients < 60 kg and 12 mg/day for those ≥ 60 kg. The intravenous administration of ATZ/BV was performed every 3 weeks at 1200 mg of ATZ plus 15 mg/kg body weight of BV.

In the event of drug-related toxicity, dose modifications or treatment delays were permitted until the resolution of symptoms or the return of the patient’s condition to baseline. Treatments were discontinued with tumor progression, unacceptable/severe adverse events, losses to the follow-up, or the withdrawal of patient consent.

### 2.3. Clinical Outcome Assessment and Statistical Analysis

Safety and/or treatment responses were evaluated every 2 to 8 weeks. Follow-ups of medical records and outpatient visits were continued until February 2022 or the date of death. Baseline characteristics and disease factors are shown as the mean ± standard deviation, a median (range or interquartile range), and a number (%) where appropriate. Overall survival (OS) was analyzed with the Kaplan–Meier method using the log-rank test. Multivariate analyses were employed to evaluate prognostic factors using a Cox proportional hazard model which was applied in a stepwise manner (both forward and backward) to evaluate the power of achievement of SVR, BCLC stage, ALBI score, FIB-4 index, and 10 other clinical features that are known predictors of survival. Analyses were performed using SPSS software (version 28; SPSS, Chicago, IL, USA). Two-sided *p*-values < 0.05 were considered to be significant.

## 3. Results

### 3.1. Baseline Characteristics

A total of 4478 HCC patients received systemic chemotherapy (*n* = 1098), transcatheter arterial chemoembolization/hepatic arterial infusion chemotherapy (*n* = 1348), or thermal ablation, including radiofrequency ablation or microwave ablation (*n* = 1770), or underwent surgical resection (*n* = 261) at Kindai University Hospital. A total of 359 patients with HCV-related HCC who received first-line systemic therapy were ultimately included (Figure 1). The baseline features of the study population are shown in Table 1. There were 277 (77%) men and 82 (23%) women, with a median age of 74 years at enrollment. According to the Barcelona Clinic Liver Cancer (BCLC) staging system, 157 patients (43.7%) were assigned to the intermediate stage (B) and 202 (56.3%) to the advanced (C) stage. The median FIB-4 index was 5.50, serum albumin 3.5, total bilirubin 0.80, alanine aminotransferase (ALT) 35 and ALBI score −2.23. Patients were also more likely to have a fibrosis-4 (FIB-4) index > 3.25, Child–Pugh class A, and higher serum levels of alpha-fetoprotein and Des-γ-carboxy prothrombin (DCP). They were treated with sorafenib (*n* = 275), lenvatinib (*n* = 48), or ATZ/BV (*n* = 36) as first-line therapy for advanced HCC.

### 3.2. Survival Analysis

The median follow-up duration was 16.0 months (range, 1.0–115.7) and the median duration of first-line systemic therapy was 3.7 months (range, 0.7–86.9). Median OS was 21.5 months (95% CI: 13.2–39.7) for the entire study cohort, with survival being significantly longer in the post-SVR HCC group than in the viremic HCC group (median, 38.4 [95% CI: 16.5–60.3] vs. 14.3 months [95% CI: 10.9–17.7], *p* < 0.01) (Figure 2a). In a subanalysis of the BCLC stage, the survival of patients with BCLC stage A/B or C significantly differed between the two groups (median, 21.7 [95% CI: 16.5–26.9] vs. 11.9 months [95% CI: 8.8–15.0], *p* = 0.05) (Figure 2b). In addition, each elevated AFP (> 90 ng/mL) was identified as prognostic factors of OS (median, 25.8 [95% CI: 20.4–31.2] vs. 10.6 months [95% CI: 8.1–13.1], *p* < 0.001) and elevated DCP (> 500 mAU/mL) was also (median, 24.7 [95% CI: 19.2–30.2] vs. 12.9 months [95% CI: 10.3–15.4], *p* < 0.001). The ALBI grade correlated with OS (*p* < 0.001), with a median OS of 19.4 months [95% CI: 11.6–27.2] for ALBI grade I patients (*n* = 75) vs. 14.4 months [95% CI: 10.2–18.5] for ALBI grade II patients (*n* = 181) and 4.3 months [95% CI: 3.1–5.6] for ALBI grade III patients (*n* = 13) (Figure 2c). OS was significantly longer in HCV-related HCC patients with a FIB-4 index < 4.0 than in those with a higher FIB-4 index (median, 19.9 [95% CI: 10.1–29.8] vs. 11.9 months [95% CI: 8.6–15.2], *p*  =  0.003) (Figure 2d).

### 3.3. Independent Factors for OS of HCV-Related HCC Patients after Molecular Targeted Chemotherapy

Based on the multivariate Cox proportional hazards model, BCLC stage C (hazard ratio [HR], 1.816, 95% CI 1.331–2.477, *p*  <  0.001), ALBI grade II/III (HR, 1.484, 95% CI 1.060–2.079, *p*  =  0.022), extrahepatic spread (HR, 1.976, 95% CI 1.405–2.780, *p* < 0.001), baseline AFP level ≥ 90 (HR, 1.433, 95% CI 1.021–2.012, *p* = 0.038), baseline DCP level ≥ 500 (HR, 1.605, 95% CI 1.135–2.270, *p* < 0.001), a FIB-4 index ≥ 4 (HR, 1.522, 95% CI 1.079–2.146, *p*  =  0.017), and the achievement of SVR (HR, 0.464, 95% CI 0.294–0.732, *p*  =  0.001) were independent risk factors for HCV-related HCC (Table 2). Meanwhile, the optimal cut-point values such as age 70, ALT 27, FIB-4 4.0, AFP 90, and DCP 500 were defined as the value whose sensitivity and specificity are the closest to the value of the area under the ROC curve. However, age, sex, baseline ALT levels, and vascular invasion were not independent risk factors for OS.

### 3.4. Dynamic Changes in the Liver Abnormality during First Chemotherapy

Figure 3 shows dynamic ALBI score changes from pre-treatment to the end of first chemotherapy of the patients between SVR and HCV viremia. A low level of liver abnormality remained in patients who had achieved SVR; however, there was no difference in ALBI score within three months. Importantly, the SVR worked better to differentiate the risk of ALBI score deterioration at the end of the first molecular targeted therapy (*p* = 0.034).

### 3.5. Second and Subsequent Chemotherapies

Of the 359 patients with HCV-related HCC receiving first-line treatment, 60 also received second-line treatment, 33 received a third-line treatment, 14 received a fourth-line treatment and 8 received a fifth-line treatment. Second-line chemotherapy consisted of lenvatinib (*n* = 39), ATZ/BV (*n* = 11), sorafenib (*n* = 7), and regorafenib (*n* = 3). ATZ/BV, sorafenib, and ramucirumab were the most-prescribed third-line regimens. Third or later-line treatment was allowed to deliver two or more previous regimens as rechallenge. Although most HCC patients with HCV viremia did not receive subsequent chemotherapy, a total of 56 patients (35.7%) who had achieved SVR could receive second- or later-line chemotherapy in the SVR-achieved group (*p* < 0.001) (Figure 4).

## 4. Discussion

The present study investigated the prognostic factors for survival in patients with HCV-related HCC after molecular targeted chemotherapy. The obtained results revealed that SVR, the BCLC stage, ALBI grade, and FIB-4 index were independent prognostic factors for OS. As expected, SVR was most strongly associated with OS. The progression of HCV-related liver disorders is associated with continuous viral replication and may cause fibrosis and cirrhosis. The first goal of anti-HCV therapy is to reduce liver inflammation, and anti-HCV therapy may improve liver function not only in patients with chronic hepatitis C, but also in those who ultimately develop HCC. Meta-analyses revealed > 70% reductions in the incidence of HCC and a 4.6% decrease in absolute risk after SVR [19], and the recurrence of HCC after radical therapy was also expected to be reduced by the successful treatment of HCV [20]. Previous studies concluded that HCC patients who achieved SVR had longer OS than those who did not, or those without antiviral treatment [21,22,23,24]. Furthermore, Luo et al. reported that SVR was the only factor that correlated with OS in patients with HCV-related HCC in a multivariate analysis [21].

The present study also confirmed that the BCLC stage was a strong prognostic factor for patients with HCV-related HCC. Previous survival analyses revealed the prognostic significance of age, sex, marital status, tumor size, vascular invasion, and stage at diagnosis in HCC patients [25,26,27,28,29]. The BCLC staging system consists of individual components, such as performance status, microvascular invasion, extrahepatic spread, and tumor burden. Therefore, the BCLC stage may be regarded as a useful predictor of outcomes in clinical practice because it includes microvascular invasion and the underlying conditions of patients. The measurement of serum levels of tumor markers for HCC is also an important tool for disease management because they generally become elevated as tumors increase in size and number and ultimately progress to portal vein invasion [30,31,32,33]. Although in clinical practice tumor marker levels sometimes reflect tumor burden accurately and sometimes not, previous studies reported that an increase in tumor marker levels indicated a high degree of HCC malignancy regardless of morphological progression [33,34]. In our population, high values for tumor markers at the time of molecular targeted first-line therapy were also an independent predictor of OS.

The disease outcomes of patients with HCC after treatment were partly affected by the existing physiological status of the liver. In the present study, the ALBI score and FIB-4 index accurately predicted the survival of patients with HCV-related HCC following the completion of molecular targeted therapy. The ALBI score has recently been employed to assess the hepatic functional reserve, and uses two objective serological markers, albumin and bilirubin, but not subjective factors, including ascites and encephalopathy, in contrast to the Child–Pugh grade [35]. Therefore, patients with the same Child–Pugh grade may be classified into different ALBI grades and show survival differences with a wide hepatic functional reserve range within a single Child–Pugh classification, which were significantly stratified for OS in the present study. Although age, AST or ALT levels, and the platelet count have been associated with hepatic fibrosis [36,37,38,39], the FIB-4 index is also a non-invasive tool for assessing hepatic fibrosis [40]. Recent studies suggested that the FIB-4 index functions as a predictor of outcomes in patients with chronic liver disease [41,42,43]. Liver damage has been identified as a severe adverse event of targeted therapy. Previous studies that investigated the toxicity of individual molecularly targeted drugs revealed that drug-induced liver damage may result in hepatic vein thrombosis, chronic hepatitis, steatosis with an inflammatory reaction, and ultimately liver fibrosis [44,45,46]. Regarding liver monitoring, repeated non-invasive tests of liver fibrosis, such as the FIB-4 index, may facilitate changes to the management of patients with HCC treated with systemic chemotherapy.

A wide variety of anti-cancer drugs exhibit hepatotoxicity, which must be monitored because proper strategies such as discontinuation or dose modification is required. Tyrosine kinase inhibitors (TKIs) and vascular endothelial growth factor (VEGF) inhibitors are designed to target specific signaling molecules or cell receptors to block oncogenic pathways such as angiogenesis, growth signaling, and cell-cycle amplification, and allow for patient-tailored treatment based on the mutational profile of their cancer [47,48]. ICIs block regulatory pathways that normally attenuate immune function and thus disinhibit immune cells to destroy cancer cells. Hepatotoxicity can be caused by damage to structures such as the liver sinusoids, vasculature, bile ducts, and direct damage to hepatocytes themselves; moreover, occlusion of vascular and ductal structures, toxic metabolite formation, and inflammatory cell infiltration into the liver parenchyma can induce damage [49]. In our cohort, patients who achieved SVR had a substantially lower risk of poor liver function during primary systemic chemotherapy. Additionally, many patients with SVR could receive second- or later-line therapies during their HCC treatment. Chemotherapy-associated hepatotoxicity may have more relevance in patients with underlying liver cirrhosis. Our results suggest that patients whose liver function was never controlled by previous treatments may not be good candidates for conventional later-line treatment.

Some other confounding factors may influence clinical outcomes. Statins are widely prescribed to reduce cholesterol levels and have been used for the prevention and treatment of various cardiovascular diseases. Meta-analysis revealed that statins significantly reduced the risk of HCC in CHB or CHC patients (RR = 0.47; 95% CI = 0.38–0.56; *I^2^* = 77.2%) [50]. Some studies have demonstrated that statins inhibit fibrogenic hepatic stellate cell activation by nitric oxide synthase [51]. Aspirin, also known as acetylsalicylic acid, is a widely used anti-inflammatory, and epidemiological evidence and clinical experiments have found that aspirin also plays an important role in tumor prevention. Aspirin users were less likely to develop HCC than nonusers [adjusted odds ratio (OR), 0.54; 95% confidence interval (CI): 0.44–0.66], and aspirin has protective effects against HCC on people with chronic liver disease [OR, 0.46; 95% CI: 0.31–0.67] [52]. Angiotensin converting enzyme (ACE) inhibitors or angiotensin receptor blockers (ARBs) are used as first-line drugs for the management of hypertension. Inhibition of the renin-angiotensin aldosterone system has been demonstrated to reduce fibrogenesis in various organs, including the liver [53]. ACE inhibitors or ARBs for treatment of hypertension had a negative association with hepatic fibrosis (OR, 0.37; 95% CI, 0.21–0.65; *p* = 0.001) [54]. The cumulative HCC recurrence rate was lower in the ACE inhibitor-treated patients than in untreated patients (~40% vs. 75% recurrence [*p* < 0.01]) [55].

The present study had several limitations. Conclusions were based on a retrospective analysis of data from a single liver center and were weakened by the small sample size; therefore, a selection bias may exist. Second, some patients had replaced chemotherapy as the first-line treatment during the follow-up period, and the decision of which agent to be used could depend on previous exposure to chemotherapeutic agents (not to be used again if possible). We did not evaluate indicators such as dose intensity, number of cycles of chemotherapy, treatment sequencing, and toxicity, which might be related to the therapeutic effect. Unfortunately, analyses of the influences of treatment after recurrence and SVR achievement before or after HCC development on survival was not feasible with our data. In addition, we could not evaluate the effect of statins, aspirin, ACE inhibitors, or ARBs.

## 5. Conclusions

We demonstrated that SVR, BCLC stage, ALBI grade, and the FIB-4 index were associated with improved OS in patients with HCV-related HCC after molecular targeted chemotherapy. Essentially, the achievement of SVR was the strongest factor. Other factors, such as the BCLC stage, extrahepatic spread, baseline tumor marker (AFP/DCP) levels, ALBI grade, and FIB-4 index should be considered in the management of patients with HCV-related HCC. Tumor burden and severity of underlying liver disease have a significant impact on survival after first-line chemotherapy. Patients presenting with impaired liver function may need to be evaluated for other treatment or receive closer oncological follow-up.

## Figures and Tables

**Figure 1 cancers-14-04850-f001:**
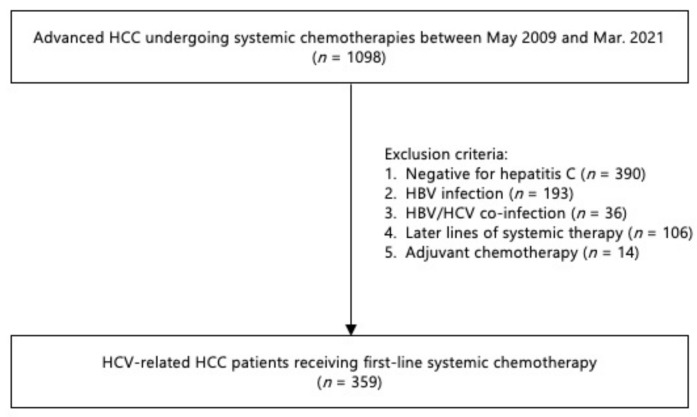
Schematic flowchart of the enrollment process.

**Figure 2 cancers-14-04850-f002:**
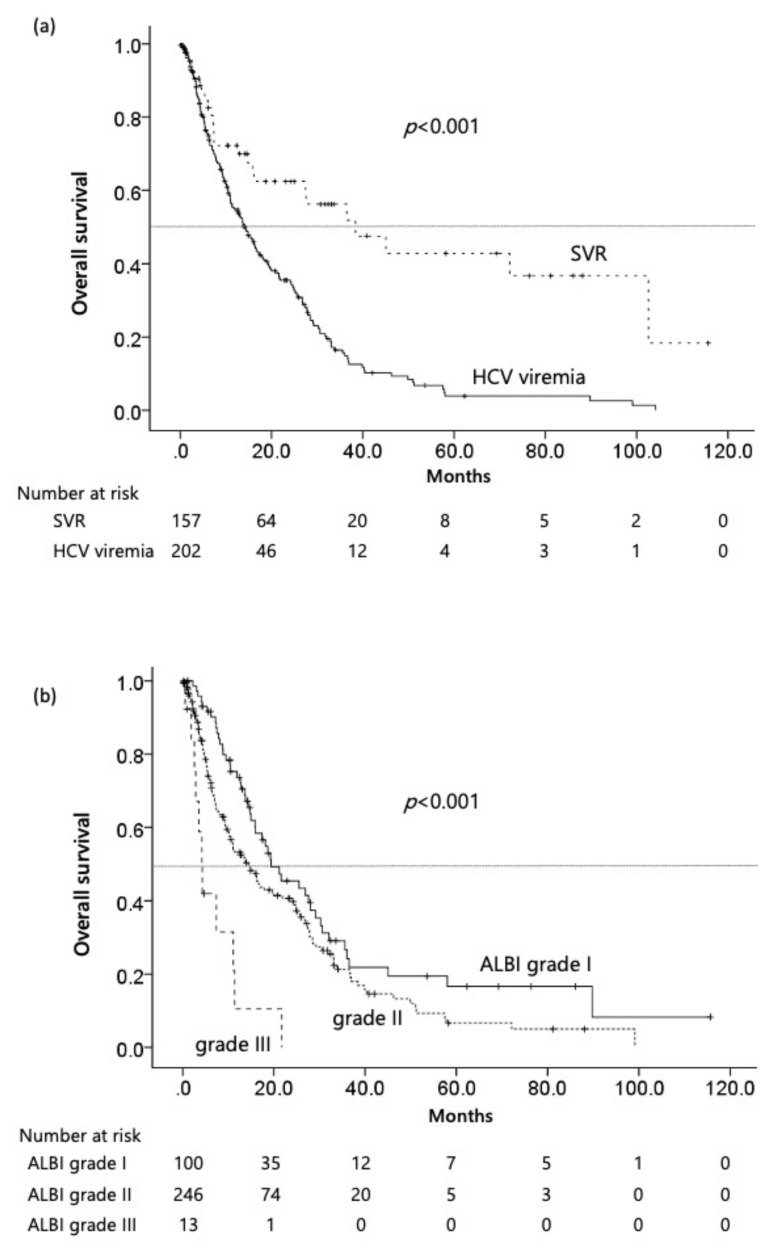
Kaplan–Meier curves of OS in HCV-related HCC patients treated with first-line molecular targeted therapies. (**a**) SVR vs. HCV viremia; (**b**) ALBI grade I and II vs. III; (**c**) BCLC stage I/II vs. III; (**d**) FIB-4 index < 4 vs. ≥ 4.

**Figure 3 cancers-14-04850-f003:**
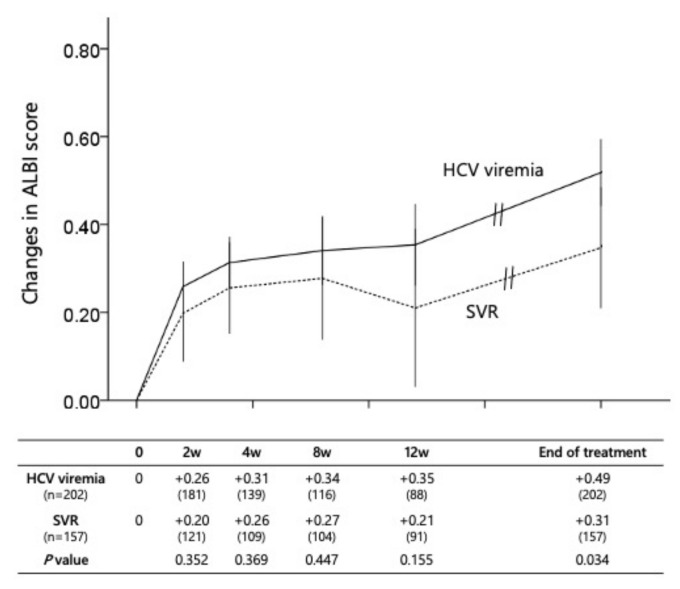
Dynamics of ALBI score during first systemic chemotherapy: SVR vs. HCV viremia.

**Figure 4 cancers-14-04850-f004:**
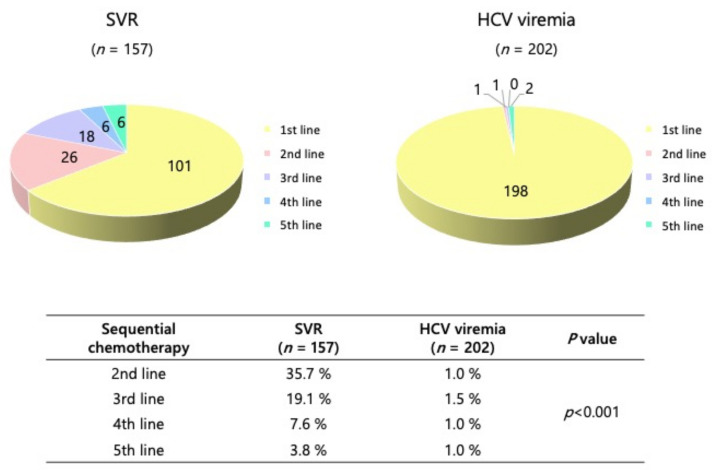
Sequential chemotherapies in HCV-related HCC patients treated with first-line molecular targeted therapies: SVR vs. HCV viremia. A total of 101, 56, 30, 12, and 6 patients received first-, second-, third-, fourth- and fifth-line chemotherapy in the SVR group, respectively. On the other hand, just 4 patients could receive second- or later-line treatment in the HCV viremia group.

**Table 1 cancers-14-04850-t001:** Baseline Characteristics of Patients.

Sex	
Male/Female	277/82
**Age (year)**	
Median (range)	74 (42–94)
**Body mass index**	
Median (range)	22.2 (13.1–32.0)
**BCLC stage C, %**	56.3
**Vascular invasion, %**	29.8
**Extrahepatic spread, %**	33.4
**FIB-4 index**	
Median (range)	5.50 (0.81–28.9)
**Serum albumin (g/dl)**	
Median (range)	3.5 (1.9–5.1)
**Serum total bilirubin (g/dl)**	
Median (range)	0.80 (0.2–2.5)
**Serum ALT (IU/mL)**	
Median (range)	35 (8–232)
**ALBI score**	
Median (IQR)	−2.23 (−3.55 to −1.10)
**Child–Pugh class A, %**	75.2
**Serum AFP (ng/mL)**	
Median (range)	148.5 (1.0–1,523,200)
**Serum DCP**	
Median (range)	687.0 (7.0–1,805,900)
**First-line chemotherapy regimen**	
Sorafenib/Lenvatinib/ATZ/BV	275/48/36

BCLC Barcelona Clinic Liver Cancer, FIB-4 fibrosis-4, ALT alanine aminotransferase, ALBI albumin-bilirubin, IQR interquartile range, AFP alpha-fetoprotein, DCP Des-γ-carboxy prothrombin, ATZ/BV the combination of atezolizumab and bevacizumab.

**Table 2 cancers-14-04850-t002:** Prognostic factors for OS in HCV-related HCC patients after molecular targeted chemotherapy.

Factor	Category	Hazard Ratio	95% CI	*p* Value
Age	<70	1		
	≥70	1.150	0.83–1.592	NS
Sex	Female	1		
	Male	1.009	0.715–1.424	NS
BCLC stage	stage A/B	1		
	sage C	1.816	1.331–2.477	<0.001
Extrahepatic spread	No	1		
	Yes	1.976	1.405–2.780	<0.001
Vascular invasion	No	1		
	Yes	1.130	0.771–1.657	0.530
AFP	<90	1		
	90≥	1.433	1.021–2.012	0.038
DCP	<500	1		
	≥500	1.605	1.135–2.270	<0.001
ALT	<27 IU/l	1		
	≥27 IU/l	1.201	0.859–1.680	NS
ALBI grade	grade I	1		
	grade II/III	1.484	1.060–2.079	0.022
FIB-4 index	<4.00	1		
	≥4.00	1.522	1.079–2.146	0.017
Achievement of SVR	No	1		
	Yes	0.464	0.294–0.732	0.001

SVR sustained virological response, NS not significant.

## Data Availability

The data presented in the present study are available upon request from the corresponding author.

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
