# Peer review of "Prognostic Factors for Overall Survival in Patients with HCV-Related HCC Undergoing Molecular Targeted Therapies: Beyond a Sustained Virological Response"

_cancers, 2022, doi:10.3390/cancers14194850_

Round 1
Reviewer 1 Report
This interesting paper performs a retrospective analysis of prognostic factors for OS in 359 HCV-related HCC patients after molecular targeted systemic treatment. The achievement of a sustained virological response (SVR) (p < 0.001), albumin-bilirubin (ALBI) grade II/III (p < 0.001), Barcelona Clinic Liver Cancer (BCLC) stage C (p = 0.005), and a fibrosis-4 (FIB-4) index ≥4 (p = 0.003) were identified as prognostic factors for overall survival. Although the study has several limitations due to its retrospective nature ,the findings are clinically meaningful. Unfortunatelly ,many confounding factors that may influence the results are not possible to analyse , such as the effect of concomitant diabetes and associated NAFLD , use of statins, ACEinh ,antiplatelet drugs etc.
I would like to see as a remark this discussion in a brief paragraph or sendence in the limitations. Refs in the link https://pubmed.ncbi.nlm.nih.gov/?sort=date&size=100&linkname=pubmed_pubmed&from_uid=23460056
For ACEinh recent paper in Hepatology with an editorial Hepatology. 2022;76:295–297.
Author Response
- This interesting paper performs a retrospective analysis of prognostic factors for OS in 359 HCV-related HCC patients after molecular targeted systemic treatment. The achievement of a sustained virological response (SVR) (p < 0.001), albumin-bilirubin (ALBI) grade II/III (p < 0.001), Barcelona Clinic Liver Cancer (BCLC) stage C (p = 0.005), and a fibrosis-4 (FIB-4) index ≥4 (p = 0.003) were identified as prognostic factors for overall survival. Although the study has several limitations due to its retrospective nature, the findings are clinically meaningful. Unfortunately, many confounding factors that may influence the results are not possible to analyse, such as the effect of concomitant diabetes and associated NAFLD, use of statins, ACEinh, antiplatelet drugs etc.
I would like to see as a remark this discussion in a brief paragraph or sentence in the limitations. Refs in the link https://pubmed.ncbi.nlm.nih.gov/?sort=date&size=100&linkname=pubmed_pubmed&from_uid=23460056
For ACEinh recent paper in Hepatology with an editorial Hepatology. 2022;76:295–297.
We added a brief paragraph in the section of Discussion [Page 10, line 13-30] and a sentence in the study limitations [Page 10, line 39-40].

Reviewer 2 Report
The authors investigated the prognostic factors in HCV-related HCC patients who underwent systemic treatment as first-line therapy. The idea is of interest in its field, the methods sound adequate, and the manuscript is well-prepared. However, I am afraid that the findings do not add to the already available evidence.
Below please find my further comments:
- According to line 89, “Inclusion criteria were as follows: an age of 20-79 years”. However, the age range is reported to be 42-94 in Table 1.
- The exclusion criteria in Figure 1 do not match the exclusion criteria in line 92.
- Why did the authors choose 70 years old for age grouping?
- Subsection 3.2 (multivariate Cox) should logically come after subsection 3.3 (univariate survival analysis).
- Lines 218-220: The authors stated that "In our population, high values for tumor markers at the time of molecular targeted first-line therapy were not an independent predictor of OS in the multivariate analysis”. This statement is not supported by the presented data (especially in Table 2).
- It should not be overlooked that analyzing patients under treatment with different agents all together as a single population is a potential source of error. I suggest analyzing subgroups of patients based on the individual agent they have received.
Author Response
- The authors investigated the prognostic factors in HCV-related HCC patients who underwent systemic treatment as first-line therapy. The idea is of interest in its field, the methods sound adequate, and the manuscript is well-prepared. However, I am afraid that the findings do not add to the already available evidence.
Below please find my further comments
- According to line 89, “Inclusion criteria were as follows: an age of 20-79 years”. However, the age range is reported to be 42-94 in Table 1.
We apologize for the mistake. The inclusion criteria were revised. [Page 3, line 2]
- The exclusion criteria in Figure 1 do not match the exclusion criteria in line 92.
The exclusion criteria were revised. [Page 3, line 5-6]
- Why did the authors choose 70 years old for age grouping?
The median age of our population was 74 years old, and the optimal cut-point value such as 70 years old was defined as the value whose sensitivity and specificity are the closest to the value of the area under the ROC curve.
- Subsection 3.2 (multivariate Cox) should logically come after subsection 3.3 (univariate survival analysis).
The subsection 3.2 moved after the subsection 3.3.
- Lines 218-220: The authors stated that "In our population, high values for tumor markers at the time of molecular targeted first-line therapy were not an independent predictor of OS in the multivariate analysis”. This statement is not supported by the presented data (especially in Table 2).
When we evaluated them again, each elevated AFP and DCP was identified as prognostic factors of OS. We are sorry for the confusion.
We added a comment in the section of Results [Page 5, line 9-12] and revised a sentence in the section of Discussion [Page 9, line 25-29].
- It should not be overlooked that analyzing patients under treatment with different agents all together as a single population is a potential source of error. I suggest analyzing subgroups of patients based on the individual agent they have received.
Thank you for a constructive opinion. We agree that the subgroup analysis by the individual agent can certainly be meaningful.
Unfortunately, we just enrolled a relatively small number of patients who received Lenvatinib or ATZ/BV. Additionally, the follow-up period is relatively short in these patients. First, there could be insufficient numbers of patients for analyses. Second, ATZ/BV treatment is now recommended as the first line systemic therapy; however, we have had a clinical impression that the question, “How long and/or later line chemotherapy did the patients receive?” is more important on OS. Therefore, we did not report the subgroup analyses of the individual chemo agent use.
Thank you for your understanding.

Round 2
Reviewer 2 Report
Thank you for preparing the revised version. I am afraid that a single important minor point has remained:
The authors stated in lines 250-252 that higher levels of tumor markers were independent predictors of OS. I think this “independent” is a typo. If not, please include tumor markers in subsection 3.3/Table 2.
Author Response
Response to Reviewer #2
We wish to express our appreciation to the reviewer for providing insightful comments on our paper. The comments have helped us significantly improve the paper.
- The authors stated in lines 250-252 that higher levels of tumor markers were independent predictors of OS. I think this “independent” is a typo. If not, please include tumor markers in subsection 3.3/Table 2.
We added them in the subsection 3.3. and Table 2.
